# Biopolymers Produced by *Sphingomonas* Strains and Their Potential Applications in Petroleum Production

**DOI:** 10.3390/polym14091920

**Published:** 2022-05-09

**Authors:** Haolin Huang, Junzhang Lin, Weidong Wang, Shuang Li

**Affiliations:** 1College of Biotechnology and Pharmaceutical Engineering, Jiangpu Campus, Nanjing Tech University, Nanjing 211816, China; 202061118033@njtech.edu.cn; 2Research Institute of Petroleum Engineering and Technology, Shengli Oilfield Company, Sinopec, Dongying 257000, China; linjunzhang.slyt@sinopec.com (J.L.); wangweidong168.slyt@sinopec.com (W.W.)

**Keywords:** sphingomonas, sphingan, biopolymer, EOR, welan, diutan, gellan

## Abstract

The genus *Sphingomonas* was established by Yabuuchi et al. in 1990, and has attracted much attention in recent years due to its unique ability to degrade environmental pollutants. Some *Sphingomonas* species can secrete high-molecular-weight extracellular polymers called sphingans, most of which are acidic heteropolysaccharides. Typical sphingans include welan gum, gellan gum, and diutan gum. Most sphingans have a typical, conserved main chain structure, and differences of side chain groups lead to different rheological characteristics, such as shear thinning, temperature or salt resistance, and viscoelasticity. In petroleum production applications, sphingans, and their structurally modified derivatives can replace partially hydrolyzed polyacrylamide (HPAM) for enhanced oil recovery (EOR) in high-temperature and high-salt reservoirs, while also being able to replace guar gum as a fracturing fluid thickener. This paper focuses on the applications of sphingans and their derivatives in EOR.

## 1. Introduction

In 1990, Yabuuchi et al. firstly proposed the genus *Sphingomonas* based on the partial nucleotide sequence of the 16S rRNA gene, the presence of unique sphingoglycolipids, and the major type of ubiquinone [1]. The original *Pseudomonas elodea* was renamed *Sphingomonas paucimobilis*, and *Sphingomonas paucimobilis* JCM 7516^T^ was used as the model strain of this species. *Sphingomonas* belongs to the α-4 subphylum of Proteobacteria and can synthesize high molecular polysaccharide biopolymers. In earlier studies, the bacteria producing gellan and related polysaccharides were originally relegated to diverse genera, including *Pseudomonas*, *Alcaligenes*, *Azotobacter,* and *Xanthobacter*. A re-examination of the biochemical and physiological characteristics of these bacteria indicated that they were closely related to each other and *Sphingomonas paucimobilis*, previously referred to as *Pseudomonas paucimobilis* [2]. In 1993, the acidic extracellular polysaccharides secreted by members of the genus *Sphingomonas* were firstly named as sphingans by Pollock et al., with gellan as the reference material [2].

Sphingan is a general term for extracellular polymers synthesized by *Sphingomonas* strains, such as welan gum (s-130), gellan gum (s-60), rhamsan gum (s-194), diutan gum (s-657), sanxan (Ss), S-88, S-198, and S-7. As water-soluble carbohydrate polymers, sphingans have a wide range of functional properties. Like other biopolymers, they have a remarkable thickening effect, but can also be used for chelating, emulsifying, or stabilizing applications.

Enhanced oil recovery (EOR), also known as tertiary recovery, is of great value for improving the efficiency of oil extraction. One of the core technologies of tertiary recovery is the chemical flooding with surfactants and polymers as the main functional substances. Among them, hydrogel-forming polymers can increase the viscosity of fluids containing water, resulting in especially improved extraction efficiency. Among EOR polymers, chemically hydrolyzed polyacrylamide (HPAM) is the most extensively used currently [3]. However, HPAM can be degraded into toxic acrylamide monomers that can infiltrate into the groundwater [4]. By contrast, the environmental friendliness of biopolymers gives them wider application prospects than acrylamide. Biopolymers have brought wide applications in all phases of petroleum recovery, from well-drilling to wastewater treatment. Xia et al. [5] summarized the application of polysaccharide biopolymer in petroleum recovery, focusing on Xanthan gum, Scleroglucan, cellulose, Guar gum, Chitosan, and their derivatives. In recent years, the application potential of sphingans in the field of oil exploitation has also attracted increasing attention.

Schmid et al. [6] summarized the chemical structures and physicochemical characteristics of the main representatives of the sphingan family and compared the genes involved in sphingan biosynthesis. This work can guide the production of tailor-made sphingans. In this paper, we will focus on the physicochemical properties of sphingans and their potential applications in enhanced oil recovery.

## 2. Production of Sphingans

The Kelco Company first undertook substantial commercial production of Kelzan^®^ xanthan gum in early 1964. The unique and useful properties of xanthan have led to its wide acceptance as a functional additive in food, industrial, and oil-field applications. The successful commercialization of xanthan attracted further interest in microbial polysaccharides.

*Sphingomonas* species are widely distributed in the environment and can be isolated from air, water, soil, plant tissues, and even some extreme environments. Moreover, some *Sphingomonas* species are pathogenic to plants, animals, and even humans [7]. *Sphingomonas* are Gram-negative, rod-shaped, aerobic, non-spore-forming, catalase-positive bacteria, and most of them can produce a yellow pigment. The main respiratory quinone is Q-10, a benzoquinone with 10 branched isoprenoid units. Cells contain 2-OH fatty acids but lack 3-OH fatty acids. The G + C content of genomic DNA ranges from 61.0 to 72.2% [8]. All *Sphingomonas* cell membranes contain glycosphingolipids instead of lipopolysaccharide. Glycosphingolipids are composed of dihydrosphingosine, a 2-hydroxy fatty acid, and glucuronic acid. This is the key feature that distinguishes *Sphingomonas* from other Gram-negative bacteria. According to phylogenetic analysis, polyamine pattern, and fatty acid profile, the genus *Sphingomonas* is subdivided into five subgenera, including *Sphingomonas* sensu stricto, *Sphingorhabdus*, *Sphingobium*, *Novosphingobium*, and *Sphingopyxis* [9]. At present, the 10 genera related to *Sphingomonas* are *Zymomonas*, *Sphingosinicella*, *Sphingopyxis*, *Sphingomonas*, *Sphingobium*, *Sandarakinorhabdus*, *Sandaracinobacter*, *Novosphingobium*, *Erythromonas*, and *Blastomonas* [10].

Generally, the biopolymer sphingan producers belong to the subgenus *Sphingomonas* sensu stricto, and reports from the other four subgenera are rare. The species *Sphingomonas paucimobilis* is an important microbial resource for commercial biopolymer production, including *Sphingomonas* sp. ATCC 31555, which produces welan (S-130), *Sphingomonas* sp. ATCC 31961, which produces rhamsan (S-194), *Sphingomonas sp*. ATCC 53159, which produces diutan (S-657), *Sphingomonas* sp. ATCC 31461, which produces gellan(S-60), as well as *Sphingomonas* sp. ATCC 31554, *Sphingomonas* sp. ATCC 31853, and *Sphingomonas* sp. ATCC 21423, which produces S-88, S-198, and S-7, respectively [6]. Since exopolysaccharide production is most obvious when bacteria are supplied with abundant sugar and minimal nitrogen; it can be expected that other *Sphingomonas* species may also have the capacity to secrete novel sphingans when cultured under the right conditions. It is also reasonable to expect that new isolates of *Sphingomonas* might provide commercially useful products with different rheological properties based on slightly different structures.

With increasing bacterial screening, more novel sphingans have been discovered and their producers have been identified as novel species, such as *Sphingomonas pituitosa* sp. nov. [11], *Sphingomonas sanxanigenens* sp. nov. [12], and *Sphingomonas cynarae* sp. nov. [13]. Many strains of the genus *Sphingomonas* have also been reported to produce unusual types of sphingans, such as *Sphingomonas* sp. CS101 [14], *Sphingomonas* sp. R1 [15], *Sphingomonas* sp. WG [16], and *Sphingomonas capsulata* ATCC 14666 [17]. These newly discovered sphingans enriched the knowledge of biodiversity, and some new sphingans also showed special functionalities. Aylan et al. [18] reported that exopolysaccharide produced by a strain of *Sphingomonas* was a potential bioemulsifier, with emulsification indexes above 70% toward gasoline, hexane, kerosene, and used frying oil; the exopolysaccharide was identified as belonging to a sphingan group while not being a gellan gum.

Although sphingans have potential applications in EOR, their widespread use was hindered by low yields and high costs. One of the key factors affecting production is the limit of oxygen and mass transfer in the later stage of fermentation caused by the high viscosity of sphingans. At present, the yield of most sphingans in the bioreactor is about 20–30 g/L (Table 1), and studies on enhancing the production of sphingans have been of particular interest. Fermentation parameters, including pH, temperature, oxygen supply, agitation speed, nitrogen sources, and fermentation strategies are all important factors in sphingan production. Taking the gellan-producing strain ATCC 31461 as an example, in order to improve the yield and performance, researchers adopted fermentation optimization methods including: (1) using grape pomace as carbon source to produce gellan gum [19]; (2) increasing aeration at low stirring rate speed to promote the synthesis of gellan gum with low-molecular-weight [20]; (3) optimizing the culture medium to obtain a maximum yield of 43.6 g/L [21]; and (4) using different carbon sources to vary the contents of acyl groups in gellan gum [22].

However, with the development of genetic technology, rational strain improvement for sphingan production has attracted increasing attention. So far, there are three ways to improve sphingan production: (1) by expressing exogenous hemoglobin genes to improve the oxygen-carrying capacity and increase the yield [29]; (2) by regulating the key genes of precursor and polysaccharide synthesis to weaken the bypass, thereby increasing the production of polysaccharides [30,31]; and (3) by knocking out genes of pigment synthesis to reduce the costs of the extraction process [32,33]. The gene clusters for biosynthesis of gellan and the schematic drawing of gellan biosynthesis were shown in Figure 1. Manjusha et al. [34] improved gellan gum production strain by strengthening the key genes, *gelQ*, *gelK*, *gelL*, and *gelB* (Figure 1), which helped increase the biomass and water holding capacity of gellan gum in calcium ion solution. Liu et al. [35] constructed a robust carotenoid-free welan gum producing strain by integrating the global transcriptional regulator gene *irrE* into the site of *crtB*, the key gene for carotenoid synthesis; the engineered strain performed better heat resistance, higher yield of welan gum, and decreased pigment content.

At present, the separation and extraction process accounts for most of the cost of biopolymer products. Ethanol precipitation is a commonly used polysaccharide extraction method, but it consumes large amounts of solvent and loses part of the polysaccharides, resulting in low yield and high extraction cost. Several new technologies were developed and applied to reduce the extraction costs, including: (1) synthesis of low-molecular-weight polysaccharides to reduce the viscosity of the fermentation broth [37]; (2) developing extraction methods such as aqueous two-phase extraction (ATPE), ultrasonic microwave synergistic extraction technology (UMSE) and pulsed electric field assisted extraction technology (PEFAE) to increase the content of polysaccharides, resulting in products with better clarity and colloidal stability [38,39]; and (3) removing the bacteria through enzymatic hydrolysis to obtain sphingans with higher purity [40].

## 3. Sphingans in EOR

Biopolymers produced by microorganisms can increase the viscosity of injected water, thereby reducing water mobility and helping to discharge oil from production wells [41]. The biopolymer xanthan has achieved great commercial success in oil-field applications, such as oil-well drilling fluids, work-over and completion, fracturing, pipeline cleaning, and enhanced oil recovery (EOR) [42]. The successful application of xanthan in petroleum production has stimulated a large number of studies on microbial polysaccharides.

Sphingans can change the rheology of aqueous solutions by thickening liquids, suspending solids, stabilizing emulsions, or forming gels [43]. They are not only non-toxic, safe, and environmentally friendly, but also have some unique physical, chemical, and rheological properties. They have high application potential in the field of EOR and are seen as the best alternative to HPAM. Different from food and chemical industry applications, the use of sphingans in EOR is a newly developed field in recent years. Biopolymers with shear-thinning properties are more commonly used in oil reservoirs, and most sphingans have this property.

So far, four types of sphingans have been reported in EOR applications, including welan, diutan, sanxan, and gellan (Table 2). Gel-based polysaccharides, such as gellan and sanxan, can accumulate in formation microchannels when injected into the reservoir, blocking formation pores and reducing permeability. Welan and diutan are the most widely studied in EOR due to their better rheological properties than Xanthan gum.

Diutan and welan have high viscosity at low shear rates, and their heat resistance allows them to maintain pseudo-plasticity even at high temperatures [52]. Compared with xanthan gum, diutan gum has a larger molecular length and molecular weight, which gives it better stability and elasticity. Under comparable LSRV (Low Shear Rate Viscosity), the loss of frictional pressure of diutan was lower [53]. These characteristics make it suitable as a drilling and spacer fluid in the oil drilling process. Although the molecular weight of welan gum is smaller than that of xanthan gum [54], Xu, L. et al. [55] speculated that the adjacent double helices in the welan gum molecule are arranged in parallel to form a more stable zipper mode, which gives it better viscoelasticity than xanthan. Further in-depth rheological research showed that the storage modulus and the loss modulus of welan gum is greater than that of xanthan gum under the same conditions, indicating that the molecular aggregation of welan gum is stronger and the resulting aggregates are more stable. This feature helps reduce the mobility and increase the efficiency of oil displacement.

### 3.1. Molecular Structure

Sphingans are mostly acidic heteropolysaccharides with similar but not identical structures, based on a generally conserved backbone composed of a tetrasaccharide building block made of rhamnose, two glucose units, and glucuronic acid (Figure 2). Nearly all of the described sphingans differ in their side chains. The main chain structure of sphingans is relatively conserved, consisting of [→4)α-L-Rha (1→3)β-D-Glc (1→4)β-D-GlcA (1→4)β-D-Glc (→1] repeats [6]. However, the structures of S-7 and sanxan are very different. Sphingan S-7 displays a different chemical structure with a 2-deoxy glucuronic acid instead of the commonly found glucuronic acid in the backbone. Sanxan is made up of a novel [→4) β-D-Man (1→4) β-D-GlcA (1→3) α-L-Rha (1→3) β-D-Glc (→1] tetrasaccharide repeat unit [47]. Although the composition of the main chain glycone of sphingans is not significantly different, the types and positions of side-chain glycones endow sphingans with structural and functional diversity, with unique physical, chemical, and rheological properties found in each sphingan [14]. Sphingans show high stability of viscosity in solution over a wide range of pH (2–12) and temperature (100–150 °C) depending on their chemical structure. The stability of sphingans to high temperatures and acid-base environments makes them suitable as viscosity-increasing agents that are stable under the harsh conditions of the oil reservoir.

### 3.2. Gellan Gum

Gellan gum has good stability, acid resistance, high temperature resistance, which forms heat-reversible gels, while also being resistant to microbial or enzymatic attack [56]. The general gelation temperature is between 20 and 50 °C, while the gel melting temperature is between 65 and 120 °C [57].

As a representative of sphingans, gellan gum has the characteristics of high transparency, good temperature resistance, acid resistance, and good compounding properties [58]. It is often used as a food additive in dairy-based foods [59]. In recent years, it has also emerged as a new material in the medical field. It can be used as a coating agent in medical materials, such as eye drops, and soft and hard capsules. Furthermore, gellan gum also has a similar structure to agar, and its heat resistance allows it to withstand repeated sterilization treatments. Therefore, gellan gum has the potential for replacing agar as a solid medium suitable for microbial growth [60].

Gellan gum has no glycone in its side chain, which makes it the simplest and most representative structure in sphingans. The tetrasaccharide repeat unit of gellan gum is [→4)α-L-Rha (1→3)β-D-Glc (1→4)β-D-GlcA (1→4)β-D-Glc (→1], it contains O-acetate and L-glycerate groups at C-6 and C-2 of–3)-β-D-Glcp-(1 [61]. The content of acetyl and glyceric acid groups is about 3%. According to the content of acetyl groups, it is divided into high- and low-acyl gellan gum [62]. The gel formed by high-acyl gellan gum is soft and elastic, while that of low-acyl gellan gum obtained by removing the acetyl group by alkali treatment has increased hardness and brittleness [63]. This is mainly due to cations present as counterions to the charged groups of the polymer, which can promote the double-helix formation of gellan gum [64]. Although this leads to gelation, the presence of L-glyceryl will eliminate the binding site between the metal ions. As a result, the cationic aggregation is reduced, and the strength and brittleness of the gel will be reduced. Similarly, the acetyl groups on the periphery of the double helix also play a role in inhibiting cation-mediated aggregation [65], giving different characteristics to gellan gums with high- and low-acyl content.

In addition to cations that occur as counterions on the charged groups of the polymer, the addition of cations to the medium also has the effect of promoting double-helix aggregation and gelation [64]. The increase in salt concentration will lead to the increase in gelling temperature and a decrease in relaxation critical exponent, but a too-high salt concentration will lead to excessive cation-mediated aggregation and reduce the gel strength. The addition of divalent cations has a stronger effect on gel formation than monovalent cations and induces a denser filling structure in the critical gel [66].

Gellan gum has good application prospects in the petroleum industry because of its unique gelling properties. In addition to conventional polymer flooding, gellan gum can also be used to plug highly permeable channels [67]. When gellan gum is injected into the reservoir, it gels in brine water without the need for additives. Brine water contains Ba^2+^, Ca^2+^, Mg^2+^, K^+^, and Na^+^, and the effects of their salts on gellan gum are in the order BaCl_2 _ >  CaCl_2_ ≈ MgCl_2_  >  KCl  >  NaCl. By contrast, polyacrylamide often requires the addition of chromium ions for cross-linking [68]. The oil displacement coefficient of gellan gum and polyacrylamide can reach 60–65%, while the water displacement coefficient is only 30–35% [69]. Practical applications also showed that the technological effectiveness of oil recovery reached up to 2000 tons of oil per 1 ton of dry gellan, which was comparable with the best results obtained using poly (acrylamide) crosslinked with chromium ions [70].

### 3.3. Welan Gum

Welan gum is a typical pseudoplastic fluid. It can maintain high viscosity at low shear rates. It has the advantages of high stability, ideal thickening properties, good suspension, and emulsification. Its aqueous solution is stable to heat, and the viscosity remains unchanged even at a high temperature of 150 °C, while still maintaining a good shear dilution effect.

The main chain structure of welan gum is similar to gellan gum, the first glucose of the tetrasaccharide repeating unit of welan gum has L-mannose or L-rhamnose side chains at a ratio of 1:2 [71], most of the tetrasaccharide fragments have acetyl and glyceric acid groups, with an acetyl content of 2.8–7.5% [72] In an aqueous solution, the welan gum molecules mainly aggregate due to the van der Waals force within the molecule and the hydrogen bonds between the side chain and the main chain [73]. The solution properties of welan gum make it suitable for improved oil recovery, as a food additive and in the medical field.

The unique shear thinning performance of welan gum makes it useful for petroleum production. Welan gum is also stable to acid and alkali, and the viscosity is unaffected in the pH range of 2–13 [74]. It has a unique wellbore purification and suspension effect, and its anti-settling performance is especially suitable for high-viscosity systems and in spacer fluid formulations [75].

When welan gum is used in polymer flooding, it mainly interacts via the zipper mode formed by the parallel arrangement of the double helix between the molecular aggregates to draw and drag the residual oil [76]. In addition to polymer flooding, welan gum can also be used to control the formation of unexpected free water on the surface of the oil well cement slurry to reduce fluid loss. This is mainly due to the carboxylate functional groups on the welan gum molecule. It has a strong affinity for the surface of minerals, but oil-well cement fluid loss additives often use a variety of external additives. The surface sites of different additives have different adsorption properties, which may cause competitive adsorption between additives, which reduces their performance [77].

Welan gum is also used as a thickener in fracturing fluid. He et al. [78] cross-linked welan gum with homemade liquid borax to form a fracturing fluid, which exhibited high elasticity, high sand carrying capacity, and low residue. This formulation has the potential to replace water-based guar gum fracturing fluid in low-permeability oil-fields. However, the residues generated by the use of this fracturing fluid will cause damage to the formation. It is necessary to further improve the fracturing fluid formulation and select appropriate additives to provide a more environmentally friendly system that is easy to degrade.

### 3.4. Diutan Gum

Diutan gum (s-657) is produced by *Sphingomonas* sp. ATCC 53159 has been produced industrially on a large scale. Similarly to welan gum, diutan gum is a typical pseudoplastic fluid, and its shear rate is inversely proportional to the viscosity of the solution Heat resistance experiments show that its viscosity is unchanged in the range of 25–150 °C, and the alkalinity of the aqueous solution has almost no effect on stability [79].

Diutan gum, as a traditional sphingan, also has the same main chain structure as gellan gum, consisting of [→4)α-L-Rha (1→3)β-D-Glc (1→4)β-D-GlcA (1→4)β-D-Glc (→1] repeats. The tetrasaccharide repeating unit of diutan gum has a dimeric L-rhamnose side chain at the first glucose, while two acetyl groups are connected to the second glucose [80]. In an aqueous solution, it has a double helix structure similar to that of xanthan gum but is more regular [81]. Consequently, it has the highest thermo-resistance, salt tolerance, and viscosity, and its unique properties make it a powerful thickener and drag reducer.

Diutan gum can also form a gel, but the formation of the gel is mainly related to the intramolecular double helix structure and is independent of cations [76]. From the perspective of conformation, the secondary structure of diutan gum is a rod-shaped double helix, and the tertiary structure is a helical complex composed of a rod-shaped double helix connected by covalent bonds. Diutan gum will unscrew only when the content of DMSO in aqueous solution exceeds 80% and the concentration of NaOH exceeds 0.3 mol/L at 100 °C [41]. Thus, diutan gum has better heat and alkali resistance. Li et al. [82] also proved that the thermal stability and salt tolerance of diutan gum is closely related to the double helix structure and that it has significant flow control ability in porous media. Santos et al. [81] found that diutan gum is a β-type rigid material. As a natural drag reducer, it has improved drag reduction ability in turbulent flow, which is independent of the Reynolds number and remains stable even after repeated heating–cooling cycles, indicating that the depolymerization of molecules does not affect the drag reduction ability.

The pseudoplastic, gel, and viscoelastic behaviors are not only related to the internal structure of the molecule, but also directly proportional to the concentration of diutan gum, but it can also maintain good viscoelasticity at a lower concentration, so it has a lower working concentration in practical applications [4]. When replacing the existing xanthan biopolymer-based mud system, diutan gum has a lower working concentration and higher temperature resistance, which reduces the degradation of biopolymers in the shearing process and lowers the maintenance costs. In addition, it can also reduce pump pressure and improve hole cleanliness [83], while also leading to less formation damage [84].

Additionally, it was also reported that the apparent viscosity and dynamic modulus of diutan gum are better than those of xanthan gum or HPAM under the same conditions, while the viscosity remained unchanged after aging for 90 days at high temperature and high salt concentration [41]. Therefore, diutan gum is expected to improve the recovery of high-temperature and high-salinity oil reservoirs. Lai et al. [52] also proved that diutan gum can recover 19.34% of the original oil in place (OOIP) at high temperature, salinity, and heterogeneity, while the OOIP recovery of xanthan gum was only 14.15%. Diutan gum can also inhibit water channeling in the high permeability layer. These phenomena may be related to the presence of cations. When Ca^+^ and Mg^+^ ions are present, the viscosity of diutan gum will increase and a gel will be formed [85].

### 3.5. Sanxan

Sanxan gum is a relatively new bacterial gum, which is synthesized by *Sphingomonas sanxanigenes* NX02^T^. The backbone structure of sanxan is unique, with a Man(1→4)D-GlcA(1→3)L-Rha(1→3)D-Glc tetrasaccharide repeat with acetyl groups on the glucose moiety. Due to this unique structure, sanxan can be extracted by acid precipitation [47], while also having other useful features, such as thickening, shear-thinning, gelling, and emulsifying properties [86]. Sanxan gum is similar to gellan gum and can be used to prepare transparent hydrogels [87]. It has a higher gel strength than low-acyl gellan gum, allowing it to be used as a gelling agent, stabilizer, and emulsifier. Sanxan has also recently been approved by the Chinese government for use in the food industry. Lu et al. [88] evaluated the feasibility of sanxan as a drug carrier and suggested that sanxan could be used as a novel material for oral delivery.

## 4. Modification of Sphingans

Good rheological properties, heat stability, and environmental friendliness are important prerequisites for the application of biopolymers in oil-fields. However, most biopolymers still exhibit poor performance under the harsh reservoir conditions of high temperature and salinity. For problems such as solubility and severe viscosity loss, even sphingans with excellent physicochemical properties and rheological behavior are no exception. Therefore, it is indispensable to improve the performance of sphingans to make them more applicable in oil reservoirs.

From the perspective of biosynthetic polymers, the structure of biopolymers can be regulated and tailor-made, and especially the substituents of polymer are relatively easy to regulate. Schmid et al. [89] regulated defined pyruvylation and acetylation patterns via genetic modification of xanthan producing strains, and obtained seven xanthan-variants; they found acetyl and pyruvoyl groups had significant effects on the rheological properties of xanthan. They summarized the synthetic gene cluster of sphingans and also proposed the idea of tailor-made sphingans. However, there are still no published studies confirming this hypothesis. Given that the molecular manipulation platform for sphingan-producing strains is not yet complete, there is still a long way to go to realize the tailor-made sphingans.

By contrast, it is much easier to change the structure of biopolymers by chemical modification. According to the current research, the methods to improve the oil displacement effect of polysaccharides are mainly based on two approaches (Table 3): (1) by modifying the structure of polysaccharides to increase their viscosity and reduce the mobility of water, thereby improving the recovery efficiency and displacement rate [3]; and (2) by blending mixtures to improve the properties of biopolymers, such as heat stability and salinity resistance, thereby increasing the recovery rate of heavy oil [90].

### 4.1. Chemical Modification

The backbone structure of most sphingans is relatively similar. Apart from the influence of the surrounding environment, the most important determinant of their performance is the composition and content of the substitution groups and functional groups. The apparent viscosity of sphingans containing sugar groups has a linear relationship with the concentration, and the apparent viscosity of sphingans containing acetyl groups has an exponential relationship with the concentration. At the same time, the presence of acetyl groups makes sphingans sensitive to temperature and salt ions [72], which greatly affects the stability of the gel structure. Therefore, the modification of the substitution groups and functional groups plays a vital role in the performance improvement, and common methods include substitution, esterification, and graft modification (Table 4).

As xanthan gum is one of the most widely studied biopolymers and has been extensively applied in EOR, it provides numerous lessons for sphingans. Gansbiller et al. [89] found that the COO- at the pyruvoyl end of xanthan gum is mainly used to increase the concentration and gel strength, but the ordered structure of xanthan gum will decrease as the content increases [96]. Conversely, the presence of acetyl groups can stabilize the ordered structure of xanthan gum, but too many acetyl groups will lead to a decrease in viscosity. Therefore, the acetyl and pyruvoyl groups of sphingans can also be modified to improve the gel strength, apparent viscosity, and structural stability.

As xanthan gum could be modified with 3-chloro-2-hydroxypropyl trimethyl ammonium chloride (CHPTAC) to obtain better heat-resistance, Arbaa’in et al. [95] used CHPTAC to modify diutan gum, replacing some of the hydroxyl groups with quaternary ammonium groups. They found that the rheological properties of the modified diutan gum were improved, which was manifested by an increase in apparent viscosity, storage modulus, and loss modulus. Then, Arbaa’in et al. [91] also used this method to modify welan gum, and found the apparent viscosity increased as the amount of CHPTAC substitution increased; the modified welan gum had higher thermal stability, increased viscosity, and better potential to be used in high-temperature oil wells.

Li et al. [92] developed two methods for carboxyl modification to improve the water solubility, viscosity, and crosslinking ability of welan gum in the borax crosslinking system. The specific effects were related to the modifiers and functional groups used, which further met the needs of welan gum as a water-based fracturing thickener in oil production.

In addition, polysaccharides can react with other substances to form copolymers. Wang et al. [90] synthesized a graft-modified copolymer (cWL-gA) through free radical copolymerization of welan gum, acrylamide, acrylic acid, and O-20. This copolymer had better temperature and salt resistance than pure polyacrylamide, and the recovery rate was also improved. At the same time, it also exhibited better anti-aging performance in high salinity.

### 4.2. Composite Polymers

Blending mixtures to improve the properties of polysaccharides is also a commonly used method. Blending mixtures usually belong to one of two types: (1) mixtures of different polysaccharides. Nurakhmetova et al. [51] prepared a gellan–xanthan mixture to make it suitable for plugging high permeability pores in reservoirs, and it can also be used as a plugging agent for polymer flooding technology. Xu et al. [55] found that the mixture of welan gum and xanthan gum (or gellan gum) can help to improve the morphology and rheological performance of the solution in the presence of inorganic salts, which is conducive to its application in EOR; (2) the mixing of polysaccharides with other substances. Xu et al. [93] compounded the precast granular gel B-PPG with diutan gum and found that the two components have a synergistic effect on the thickening of the suspension. The system exhibited better salt and temperature resistance than uncompounded diutan gum. Bashirul Haq [94] developed a surfactant-polymer (SP) formulation by blending alkyl polyglucoside with xanthan gum. On this basis, acetone and butanone were added to the new formula to further reduce surface tension and improve viscosity in core oil displacement.

## 5. Conclusions

Sphingans have been extensively studied in the past years. The most successful example is gellan gum synthesized by *Sphingomonas*, which has good gelling properties, making it suitable for applications in the food, medical, and chemical industries. In addition, sphingans, such as welan gum and diutan gum, have unique physical, chemical, and rheological properties due to the different types and positions of substituents. Both welan gum and diutan gum are pseudoplastic fluids, with good viscoelasticity and heat resistance, and both are superior to the biopolymer xanthan gum that has been widely used in the petroleum field for decades. They can be modified to give them better rheological properties, and have the potential to increase oil recovery in harsh oil reservoirs with high temperature and high salinity.

Although the exploration and structural transformation of sphingans in oil-field applications has achieved initial results, the research on sphingans is still in the laboratory stage, requiring further efforts to improve fermentation and purification methods. The most important future directions include: (1) screening new biopolymers and selecting biopolymers with low cost and superior physicochemical properties for harsh oil reservoirs; (2) improving *Sphingomonas* strains and fermentation process to increase the yield of sphingans and reduce the cost to make it cheaper in actual oil field applications; (3) improving extraction and purification methods to reduce the loss rate, exploring the possibility of direct application of fermentation broth in oil fields; and (4) developing structure-function relationship insights of sphingans-variants via genetic modification or chemical modification.

## Figures and Tables

**Figure 1 polymers-14-01920-f001:**
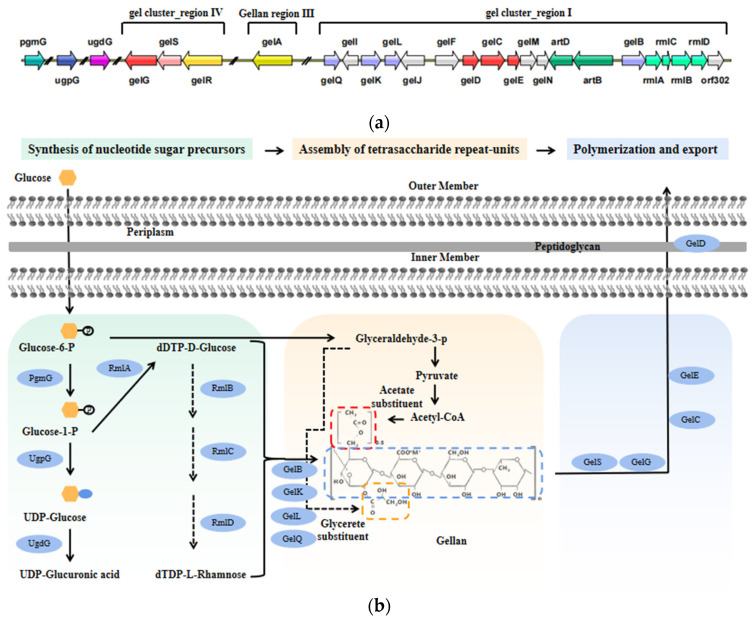
(**a**) Organization of the gellan biosynthesis gene clusters (regions I and IV) and gellan region III from *S. elodea* ATCC 31461, adapted from [36], Oxford University Press, 2002. (**b**) Schematic drawing of gellan biosynthesis, adapted from Li et al. [32], ASM Journals, 2019.

**Figure 2 polymers-14-01920-f002:**
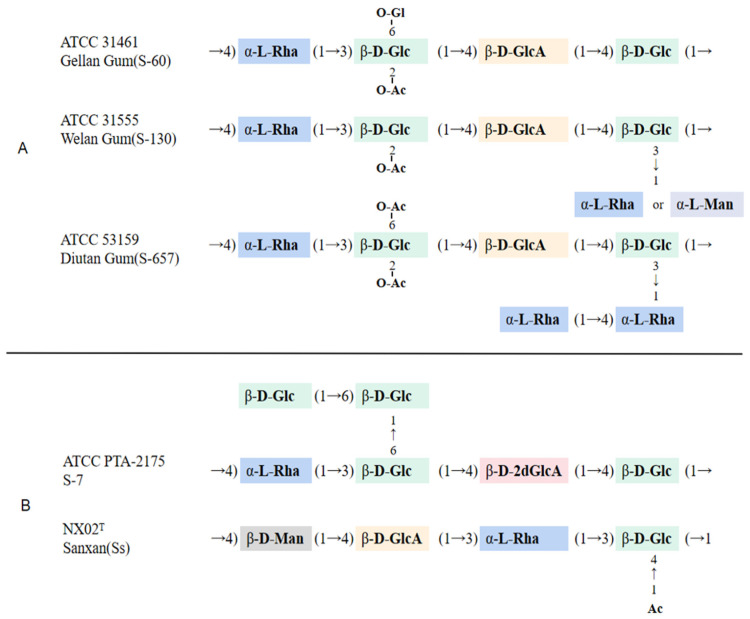
(**A**) Structures of traditional sphingans. (**B**) Structures of new reported sphingans.

**Table 1 polymers-14-01920-t001:** *Sphingomonas* strains and corresponding sphingans.

Sphingans	Strain	Bioreactor (L)	Time (h)	Yield (g/L)	Strategy	Reference
Welan	*Sphingomonas* sp. HT-1	7.5	66	22.68 ± 0.50	A two-step fermentation strategy using glucose stock solution and xylose stock solution	[23]
	*Alcaligenes* sp. CGMCC2428	7.5	72	24.90 ± 0.68	Addition of Tween-40 into the culture broth	[24]
Gellan	*Sphingomonas paucimobilis* QHZJUJW CGMCC2428	5	72	19.90 ± 0.68	A fractional factorial design was applied to investigate the main factors that affect gellan gum production	[25]
	*Sphingomonas paucimobilis* ATCC 31461	5	48	27.86	Addition of a surfactant (Triton X-100) to the medium; changing the agitation and DOT level	[26]
Sanxan	*Sphingomonas sanxanigenens* NX02	5	84	13.10 ± 0.30 (g/Kg)	Native co-utilization of glucose and xylose from corn straw total hydrolysate (CSTH)	[27]
Rhamsan	*Sphingomonas* sp. CGMCC 6833	7.5	72	21.63 ± 1.76	Using a two-stage agitation speed control strategy	[28]

**Table 2 polymers-14-01920-t002:** Molecular weight and features of several sphingans.

Sphingans	MW (Da)	Features	Reference
Welan gum	80 × 10^5^	Tolerates high concentrations of sodium and calcium ions Enhances heavy oil recovery Controls unwanted free water at the surface of the cement slurry	[44] [45] [46]
Diutan gum	72 × 10^5^	Restrains water channeling in the high-permeability layer Improves the sweep ability of the displacement phase Maintains the effect of polymer flooding for a long time.	[3]
Sanxan	4.08 × 10^5^	Absorbs onto the surface of oil droplets Forms soft, elastic thermo-reversible gels by cooling hot polysaccharide solutions	[47] [48]
Gellan gum	5.2 × 10^5^	Applied for an in-depth treatment of high and moderate permeability reservoirs. Plugging highly permeable channels in oil reservoirs and as a shut-off agent in polymer flooding technology	[49] [50] [51]

**Table 3 polymers-14-01920-t003:** Structural transformation methods and their advantages.

Modification	Specific Method	Advantages	Reference
Structural modification	The substitution of CHPTAC on the DG backbone	Increase apparent viscosity and improve heat resistance;	[91] [92]
	Carboxyethyl modification and citric acid modification	Improve water solubility, viscosity, and crosslinking ability	
	Group reaction of polysaccharides with other substances	Improve temperature and salt resistance, better anti-aging performance in high salinity	[90]
Composite polymer	Polysaccharides mixed with polysaccharides	Plugging agent for high-permeability channels in oil reservoirs; Control the morphology and rheological properties of the solution in the presence of inorganic salts	[51] [55]
	Polysaccharides mixed with other substances	Improved stability in high temperature and high salt environments; Reduce surface tension and improve viscosity in core oil displacement	[93] [94]

**Table 4 polymers-14-01920-t004:** Chemical modification methods and their advantages.

Sphingans	Functional Groups	Reaction Type	Materials	Conditions	Degree of Modification	Reference
Diutan gum	Hydroxyl groups	Substitution	CHPTAC	70 °C, 3 h	-	[95]
Welan gum	Hydroxyl groups	Substitution	CHPTAC	70 °C, 3 h	-	[91]
	Hydroxyl groups	Substitution	3-Chloropropanoic acid	5 h	0.58	[92]
		Esterification	Citric acid	80 °C, 10 h	-	
	Free-radical	Grafting	Acrylamide (AM), acrylic acid (AA), and hydrophobic monomers (O-20)	40 °C, 40 h	73.4%	[90]

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
