# Peer review of "Biopolymers Produced by Sphingomonas Strains and Their Potential Applications in Petroleum Production"

_polymers, 2022, doi:10.3390/polym14091920_

Round 1

Reviewer 1 Report

The review entitled “Biopolymers produced by Sphingomonas strains and their potential applications in petroleum production” offers a panoramic view on sphingans, biopolymers produced by different strains of Sphingomonas having useful properties exploitable for enhanced oil recovery (EOR). The manuscript describes the production of sphingans, the properties of the main components of the class as well as their applications in EOR. The final section is finally dedicated to modifications of shpingans aimed to improve their chemical-physical properties for a more performing use in petroleum production.

The topic is interesting as it offers a good start point to enrich the research on biopolymer-based sustainable technologies. The manuscript is also clear and pleasant to read.

Nevertheless, the reviewer suggests some modification in order to improve the value of the manuscript:

  • In the sections 3.3 e 3.4 there are 2 references with formatting errors. Please, edit the format.
  • Within the section 3.5 it is not well specified the possible use of sanxan gum in petroleum production. Please, specify this point, possibly adding some examples (if any).
  • All of the sphingans described are proposed for petroleum production. The text points up clearly the main properties that make sphingans potential tools in this field. However, for a deeper understanding, the addition of a new paragraph focused on a comparison between the depicted polysaccharides could be helpful to better figure out the differences between the sphingans’ possible performances in EOR and/or their relationship.
  • Figure 2 depicts the repeating units of the main sphingans. Although the monosaccharides involved are well known, it is recommended to illustrate the whole chemical structures. This could better match the journal standards as well as improved the clarity of the section 4.1.
  • Section 4.1 is very interesting from the polymer chemistry point of view, as it highlights the possibility to tailor the polymer properties as a function of the chemical groups involved. Although not mandatory, it could be interesting to enhance this section by adding further examples of sphingans’ chemical modification (or other polysaccharides potentially useful in EOR) or, alternatively, explain some details of few of the chemical modification depicted (e.g. solvents, reagents, reaction parameters).

Author Response

1.In the sections 3.3 e 3.4 there are 2 references with formatting errors. Please, edit the format.

We are sorry for that, the Chinese citation has been translated into English.

2.Within the section 3.5 it is not well specified the possible use of sanxan gum in petroleum production. Please, specify this point, possibly adding some examples (if any).

Sanxan was a relatively new sphingan with similar gelling properties of gellan gum, it was described separately in Section 3.5, but since its application had not yet involved the oil field, only a brief explanation (line 377-380) of its application direction was given here.

3.All of the sphingans described are proposed for petroleum production. The text points up clearly the main properties that make sphingans potential tools in this field. However, for a deeper understanding, the addition of a new paragraph focused on a comparison between the depicted polysaccharides could be helpful to better figure out the differences between the sphingans’ possible performances in EOR and/or their relationship.

As suggested by the reviewer, the specific applications of these sphingans in petroleum recovery are explained in the third paragraphs (line 195-198) of Section 3.

4.Figure 2 depicts the repeating units of the main sphingans. Although the monosaccharides involved are well known, it is recommended to illustrate the whole chemical structures. This could better match the journal standards as well as improved the clarity of the section 4.1.

Thank you so much for your advice. The structure of the relevant sphingans were  described detailedly in 3.2 (line 249-251), 3.3 (line 289-292) and 3.4 (line 327-329).

5.Section 4.1 is very interesting from the polymer chemistry point of view, as it highlights the possibility to tailor the polymer properties as a function of the chemical groups involved. Although not mandatory, it could be interesting to enhance this section by adding further examples of sphingans’ chemical modification (or other polysaccharides potentially useful in EOR) or, alternatively, explain some details of few of the chemical modification depicted (e.g. solvents, reagents, reaction parameters).

This suggestion is really great. We have added a Table (Table 4) to give the details of the chemical modification depicted in Section 4.1. 

Reviewer 2 Report

This review paper covers some aspects regarding biopolymers (heteropolysaccharides) produced by Sphingomonas strains and the application of sphingans in petroleum production. The manuscript was well organized and well written. As a mini review, the manuscript has given adequate background about strains, production, structural characteristics and performance features of major four sphingans. It reads smoothly and may attract significant attention from readers in different research areas. There are two small issues, that the reviewer may suggest the authors to further address.

  1. Two references cited with a foreign language different from English in the sections of 3.3 and 3.4. However, they are not present in the reference list. It is suggested to use English.
  2.  In the section of 4.2 composite polymers of sphingans, it is suggested to add a few more examples in the petroleum production using this modification of sphingans. Are there other modifications that have been used to improve the performance of sphingans in petroleum production, such as biological, enzymatic, or physical modifications?

Author Response

1.Two references cited with a foreign language different from English in the sections of 3.3 and 3.4. However, they are not present in the reference list. It is suggested to use English.

We are sorry for the mistake.

The two Chinese citations in the areticle have been translated into English.

2.In the section of 4.2 composite polymers of sphingans, it is suggested to add a few more examples in the petroleum production using this modification of sphingans.

As suggested by the reviewer, two examples about the modification of sphingans are added to this part (line 459-462, line 466-469).

3.Are there other modifications that have been used to improve the performance of sphingans in petroleum production, such as biological, enzymatic, or physical modifications?

Physical modification is almost equal to the preparation of mixtures (Section 4.2); for the enzymatic modification, it looks like very easy to implement, but no similar research content was found in the literature.

Reviewer 3 Report

Thi review is well written and logically organized, the relevant literature in the field is cited.

Minor remarks:

Please double-check the references, at page 9 one is written in chines ideograms, please translate.

Author Response

1.Please double-check the references, at page 9 one is written in chineseideograms, please translate.

We are sorry for that, the Chinese citation has been translated into English.